# Molecularly Imprinted Polymers as Solid-Phase Microextraction Fibers for the Isolation of Selected Antibiotics from Human Plasma

**DOI:** 10.3390/ma14174886

**Published:** 2021-08-27

**Authors:** Małgorzata Szultka-Młyńska, Daria Janiszewska, Bogusław Buszewski

**Affiliations:** 1Department of Environmental Chemistry and Bioanalytics, Faculty of Chemistry, Nicolaus Copernicus University, Gagarin 7, 87-100 Torun, Poland; janiszewska_daria@doktorant.umk.pl (D.J.); bbusz@umk.pl (B.B.); 2Centre for Modern Interdisciplinary Technologies, Nicolaus Copernicus University, Wilenska 4, 87-100 Torun, Poland

**Keywords:** antibiotics, electropolymerization, human plasma, mass spectrometry, polythiophene, poly (3-methylthiophene), solid-phase microextraction

## Abstract

The aim of this study was to examine the synthesis of novel molecularly imprinted polymer (MIP)-coated polythiophene and poly(3-methylthiophene) solid-phase microextraction fibers using the direct electropolymerization method. Synthesized SPME fibers were characterized with the use of various physicochemical instrumental techniques. MIP-SPME coatings were successfully applied to carry out the selective extraction of selected antibiotic drugs (amoxicillin, cefotaxime, metronidazole) and their metabolites (amoxycilloic acid, amoxicillin diketopiperazine, desacetyl cefotaxime, 3-desacetyl cefotaxime lactone, hydroxymetronidazole). Solid-phase microextraction parameters for the simultaneous determination and identification of target compounds were optimized using the central composite design (CCD), and they accounted for 5–15 min for desorption time, 3–10 for the pH of the desorption solvent, and 30–100 μL for the volume of the desorption solvent. High-performance liquid chromatography and mass spectrometry (MS) detectors such as quadrupole time-of-flight (Q-TOF MS) and triple quadrupole (QqQ MS) were applied to determine and to identify selected antibiotic drugs and their metabolites. The MIP-coated SPME are suitable for the selective extraction of target compounds in biological samples from patients in intensive care units.

## 1. Introduction

The miniaturization of analytical techniques observed in recent years is the result of the use of the most advanced technologies. It is connected with the necessity to develop more and more sensitive and precise analytical methods [1,2]. Modern medicine has a large variety of drugs used to combat various diseases caused by bacteria. Moreover, the rapidly growing number of microbial strains resistant to particular chemotherapeutic agents forces us to conduct further research [3]. The huge diversity of samples due to the composition and aggregate state of the matrix requires the use of many techniques of the sample preparation to extract the tested compounds and to purify them prior to the determination and identification step. An important stage in the preparation of the sample for the final analysis is its isolation and/or enrichment, which consists in transferring the analytes from the primary matrix to the secondary matrix with the simultaneous removal of interfering substances (isolation) and increasing the concentration of the analytes to a level above the quantification limit of the analytical methodology (enrichment) [1,4,5]. The application of extraction techniques such as the liquid–liquid extraction (LLE), the solid-phase extraction (SPE), and the super fluid extraction (SFE) allows us to separate the analyte from the matrix, to eliminate or reduce interferences from other components, and to enrich the analyte to a level that allows for the appropriate determination. It is advantageous for the process to take place in one step. In the era of “*green chemistry*”, it is more and more difficult to justify the use of analytical methods with the use of large amounts of toxic solvents at any stage of the sample preparation. Hence, solvent-free methods are preferred [1,2,3,4,5].

One of the newest techniques applied to isolate organic compounds is the solid-phase microextraction (SPME) due to the short extraction time, the fact that no solvents are needed, and the simplicity of the analysis [6]. The SPME technique is based on the adsorption of compounds on the absorbent layer covering the fiber and introducing them directly into the sample. The SPME is applied in case of various biological matrices, such as whole blood, plasma, urine, breath gas, food, and natural products [6,7,8]. The technique represents an analytical tool for the development of simple and selective medical diagnostics from samples in vitro, ex vivo or in vivo [9,10,11]. Most applied SPME coatings are commercially available. However, in the recent years, studies have been focused on the development of fibers with coatings on metal support [12,13,14]. Among these, electroconductive polymers including polypyrrole, polythiophene and poly(3-methyltiophene) are identified as promising candidates for the selective extraction of drugs in bioanalytical applications [15,16,17,18,19,20]. However, depending on the chemical nature of the isolated compounds, fibers with different sorption properties are used. The type of microextractive fiber coating and its thickness are selected on the basis of the polarity of the extracted analytes and the size of their particles. The type of stationary phase that the SPME fiber is covered with determines the sensitivity and selectivity of the method. Imprinted polymers, mostly molecularly imprinted polymers, are increasingly used as selective sorbents [21,22,23].

The purpose of this study was to prepare new MIP-coated SPME fibers by the electrochemical polymerization approach with relevant compounds as templates. For this reason, we performed deeper physico-chemical characteristics of the obtained materials with the use of different instrumental techniques. It should be mentioned that there are no available papers to study the recognition mechanisms of amoxicillin, cefotaxime and metronidazole as templates for extraction with the use of MIP-coated SPME fibers from real samples from patients. Moreover, in this contribution, we described with the use of the SPME-LC-MS/MS approach the determination and identification of the three antibiotic drugs in the human plasma samples from patients in the intensive care unit.

## 2. Materials and Methods

### 2.1. Chemicals and Reagents

All reagents and organic solvents were of analytical grade. Thiophene (99%) and 3-methylthiophene (98%) were purchased from Sigma-Aldrich (Schnelldorf, Germany). Amoxicillin (AMOX), cefotaxime (CEF), and metronidazole (MET), as well as their metabolites amoxycilloic acid (AMA), amoxicillin diketopiperazine (AMD), desacetyl cefotaxime (CEF-dAce), 3-desacetyl cefotaxime lactone (CEF-dAceLac), and hydroxymetronidazole (MET-OH), were supplied by Novartis Pharma GmbH (Nuremberg, Germany). Water was purified with a Milli-Q Purification System (Millipore, Bedford, MA, USA). Real samples from patients (human plasma) were kindly provided by Nicolaus Copernicus University, Collegium Medicum (Torun, Poland), with the Bioethical Commission’s permission (no. 585/2017).

### 2.2. MIP-SPME Coating Preparation

MIP-SPME fibers were synthesized with the use of the electrochemical polymerization (Table 1). The preparation procedure was based on the home-made set-up system coupled with the potentiostat-galvanostat. In polymerization, 0.4 M thiophene (with CEF or MET), or 0.1 M 3-methyltiophene (with AMOX) solutions in 0.1 M tetrabutylammonium tetrafluoroborate (C_4_H_9_)_4_N(BF_4_) in ACN were used, respectively. Dynamic voltamperometry in the meaning of the Linear Sweep Voltammetry (LSV), with threshold potentials from −0.2 to +2.5 V for PTh and from −0.2 to +3.0 V for PMeTh, was applied. The electropolymerization was performed with a scan rate on 50 mV/s and 7 scans. Stainless steel (SS), Ni-Cr, used as a working electrode was conditioned with a mixture of acetonitrile and water (80:20, *v/v*) to remove the contamination and then dried at ambient temperature before each research investigation. Additionally, SS wires were cyclic-potential-scanned within the potential range −0.2–+1 V in 0.1 M (C_4_H_9_)_4_N(BF_4_) solution regarding the redox peak obtained. Afterwards, the imprinted polymers were conditioned with respect to removing the template in 0.2 M Na_2_HPO_4_ (in EtOH) in case of AMOX, or in 0.05 M NaOH (in MeOH) in case of CEF and MET applying potential cycling in a range from 0 to +1.2 V. Control SPME coatings were synthesized under the same electropolymerization conditions but without adding the antibiotic drugs to the reagent solution.

### 2.3. Apparatus and HPLC-MS Analysis Conditions

The determination and identification of selected antibiotics and their metabolites were carried out using an HPLC Agilent 1290 Series system (Agilent Technologies, Waldbronn, Germany) equipped with an ESI interface, a 6540 UHD accurate mass Q-TOF detector (Agilent Technologies, Waldbronn, Germany) and Mass Hunter software (B.04.01, Agilent Technologies, Waldbronn, Germany) for instrumental control and data collection. The chromatographic XDB-C18 1.8 μm 4.6 × 50 mm column (Agilent Technologies, Waldbronn, Germany) was maintained at 30 ± 0.5 °C. The mobile phase was composed of 0.1% HCOOH with acetonitrile (25:75, *v/v*) dosed at a flow rate of 0.4 mL/min, while the injected sample volume was 1 μL. Moreover, quadrupole time-of-flight mass spectrometric measurements (LC-MS/MS Shimadzu (Kyoto, Japan)) were performed using electrospray ion source operating in positive ion mode (ESI(+)), with the following set of operation parameters: capillary voltage (CV), 3.5 kV; octopole voltage (OV), 800 V; skimmer voltage (SV), 50 V; drying gas temperature (DGT), 295 °C; shielding gas temperature (SGT), 315 °C; fragmentor voltage (FV), 195 V.

The liquid chromatograph was equipped with two LC-30AD pumps, the SIL-30AC autosampler and the CTO-20AC thermostat (Kyoto, Japan). The electrospray ionization (ESI) was applied in the positive ion mode, due to the positive charge located in the chemical structure of target compounds. Full-scan mass spectra (MS) were recorded within the mass range of *m/z* 150–500 followed by precursor ions (MS/MS) for *m/z* 100–450 (Table 2). The following operation parameters of MS/MS were applied: the gas flow rate was 6 L/min; the nebulizer gas pressure was 40 psi; the capillary voltage was 3500 V; the fragmentor voltage was 135 V; the drying gas temperature was set at 290 °C.

The obtained results were collected with the use of the following programs: Agilent Mass Hunter software version B.04.01 and LabSolution version 5.8. Moreover, they were further processed using Microsoft Excel. The instrument operated in the extracted ion chromatogram (EIC) and product ion modes, respectively.

### 2.4. SPME—Optimization of Extraction Parameters

The optimization of the extraction with the use of the MIP-SPME of target compounds was performed with the use of the central composite design (CCD). Three independent variables of extraction on three levels were optimized using the response surface methodology (RSM). The optimized parameters were the desorption time (5–15 min), the pH of the desorption solvent (3–10), and the volume of the desorption solvent (30–100 μL). An optimization study for the three parameters was performed with the use of Statistica 12.0 (StatSoft, Tulsa, OK, USA). The whole CCD design consists of 15 factorial points.

### 2.5. Characterization of MIP-SPME Coatings

#### 2.5.1. Cyclic Voltammetry (CV)

During the electropolymerization, a home-made set-up system combined with a high-performance potentiostat/galvanostat Autolab PGSTAT128N model (Utrecht, The Netherlands) equipped with Gpes software for voltammetric measurements were applied for the synthesis.

#### 2.5.2. Scanning Electron Microscopy (SEM)

A micrograph of the SPME coating regarding the morphological evaluation was obtained through scanning with the electron microscope (SEM), LEO 1430VP (LEO Electron Microscopy, London, England) from Carl Zeiss SMT (Oberkochen, Germany). It was used for SEM imaging of SPME sorption polymer fibers deposited on a solid Ni-Cr support. Moreover, the mechanical and chemical stability was set up with the Optical Stereomicroscope model SZX16 (Olympus, Tokyo, Japan) equipped with a CCD camera (Olympus, Tokyo, Japan) and the CELL software (V1.1.6, Olympus, Tokyo, Japan).

#### 2.5.3. Infrared Spectroscopy (FTIR)

The Spectrum 2000 spectrometer from Perkin Elmer (Waltham, Winter St, MA, USA) was used for the analyses. The spectra were made having pressed the sample with potassium bromide (KBr pelleting technique).

#### 2.5.4. Small Angle X-ray Scattering (SAXS)

An SAXS low-angle powder diffractometer (Nanostar, Bruker AXS, Karlsruhe, Germany) with a copper lamp was used for the analyses. The camera was equipped with a HiSTAR proportional multi-wire detector with a resolution of 1024 × 1024. The optics of the device (focusing and monochromatization) was in the form of crossed Goebel mirrors. The room temperature was measured. The measurement time was 10,000 s, while the distance sample–detector was 650 mm.

#### 2.5.5. Transmission Electron Microscopy (TEM)

A transmission electron microscope manufactured by FEI Europe (Hillsboro, OR, USA), the Tecnai F20 X-Twin model, was used for the analysis.

#### 2.5.6. Nuclear Magnetic Resonance (NMR)

As for the polymer samples, it was necessary to dissolve them with a deuterated solvent. Each sample was suspended in 1 mL of CDCl_3_ and placed in an ultrasonic bath. The polymer fibers were shredded under the influence of ultrasound (30 min). The samples were then re-centrifuged for 10 min at 4 °C at 12,000 rpm, and 550 µL was transferred to a 5 mm NMR cuvette. NMR measurements were made using the AVANCE II spectrometer by BrukerDaltonik GmbH (Bremen, Germany), with the operating frequency of 600.58 MHz. A 90-degree pulse was used for the measurements.

## 3. Results and Discussion

### 3.1. Synthesis of MIP-SPME Coatings

The exemplary voltamperogram obtained during the electropolymerization of monomer (thiophene) is shown in Figure 1. It turns out that the intensity of reaction peaks appearing in voltamperograms during the monomer oxidation is significantly influenced by the reaction of the environment. Due to the fact that the polymerization was carried out at a pH close to neutral, these signals were not very intense. The applied potential of the electropolymerization process changes slightly during the entire synthesis. The change in the potential over the time of a unit repetition (scan) is similar in each reversion cycle. This may indicate that the course of the electrodeposition process is stable throughout the entire duration of the process and is not disturbed by the process of forming a polymer coating on the surface. Moreover, in this case, a one-stage polymerization of the monomer with a high concentration was also used at the beginning of the reaction. It is interesting that during the polymerization of the thiophene, signals from the processes taking place on the electrodes were visible. A different situation was observed in the case of 3-methylthiophene. On the voltamperogram obtained during the electropolymerization process, the current intensity at the minimum potential decreases along with the increasing number of repetitions (scans). Moreover, the current value increases at the maximum voltage value. An interesting fact is that the alkyl derivative of thiophene is much easier to oxidize compared to the pure monomer without substituents.

Additionally, multi-cyclic potentiodynamic curves for a model compounds template (AMOX, CEF, MET) were obtained under the printing conditions, in the absence and presence of functional monomers (PTh, PMeTh). When recording the potential cycles for a drug molecule in 0.1 M (C_4_H_9_)_4_N(BF_4_) acetonitrile solution, no anode peak was observed corresponding to electro-oxidation. Its absence indicates that, during the electropolymerization process, the target drug template was imprinted in the MIP layer in its unchanged form. Similar profiles were obtained for all studied drugs. Moreover, this may indicate the presence of strong interactions caused by the presence of the sulfur atom and a free electron pair in the structure of the polythiophene in case of CEF and MET. Moreover, a slight reduction in the hydrophobicity of polythiophene by introducing a methyl group in the 3-position of the thiophene ring resulted in a favorable change manifested by an increase in the extraction efficiency for AMOX. In the synthesized MIP layer, there are molecular voids filled with the imprinted template (relevant drug). Having been removed, the polymer leaves an imprint in the form of a molecular gap with binding sites. In addition, during the electropolymerization, the oxidation of the analytes (molecules to be imprinted) is disadvantageous, since this can lead to the imprinting of the oxidation products (interfering substances) instead of the desired compound. Then, they can form binding sites that are unable to interact effectively with the appropriate analyte. In turn, the interaction of interfering substances with MIP molecular cavities varies according to their structural similarity to the printed template. If the conditions for the impression step of the template molecule (drug) are properly selected, then the molecular voids of the MIP layer should be complementary in shape and size to the template molecules, not to the interfering substance molecules. The obtained results show that polymer sorption fibers with an imprinted drug (AMOX, CEF and MET) obtained during the experimental work were characterized by different selectivity. Hence, PMeTh allows for obtaining the highest extraction recovery and selectivity toward AMOX. On the other hand, PTh allows for obtaining the highest extraction recovery and selectivity toward CEF and MET.

### 3.2. Physico-Chemical Characterization of Synthesized MIP-SPME Coatings

Scanning electron microscope images made it possible to measure the thickness of the fibers synthesized by electropolymerization. Moreover, they provided information on the structure of the fibers and the individual grains of the material. Table 1 presents the values corresponding to the given types of polymer fibers. The coatings obtained under the conditions described above are characterized by dark grey, graphite color. Their surface is matte and rough. The surface morphology of the obtained materials shows the presence of deposited polymer layers. The layers obtained at the stage of impressing the molecule of a given biologically active compound (antibiotic drug) are characterized by a matt, velvety, light-gray surface, do not detach from the substrate and do not crack. Their surface is homogeneous and presumably much less developed compared to that of a “pure” polymer fiber.

On the basis of the obtained photos presented in Figure 2, a homogeneous surface of the fibers can be seen. As a result of mixing the solution during the electropolymerization process, a clearly visible boundary between the electrode surface and the obtained sorption fiber can be observed. Moreover, mixing at the speed of 300 revolutions per minute made it possible to obtain homogeneous surfaces due to the equal concentration of monomers after the oxidation of oligomers and polymers in the entire volume of the solution (Figure 2). Moreover, in the case of conducting the electropolymerization process in a small volume of the reaction solution, it is very important that the polymer precipitation process on the electrode is not disturbed by mixing the solution. As a result of the conducted mechanical tests, it was found that the coatings adhered well to the substrate and showed no cracks or internal stresses.

The attached photos confirm the effectiveness of the electrodesorption process. In the case of the photos of the fiber before this process, clearly brighter areas are visible. This is due to the use of a secondary electron detector that provides information only on the sample surface topography. The use of a backscattered electron detector allows for an initial analysis of the composition of the sample by detecting the contrast between places on the surface of the sample that differ in their chemical composition.

The EDX spectrum shows that the signals from the individual elements on the surface of the SPME coating with the drug molecule imprinted on it are higher than the signals from the layer without the analyte molecule imprinted. The experiments were performed after removing the sorption material (PTh, PMeTh) from the working electrode (SS, Ni-Cr). Elements with higher atomic masses are definitely brighter in the SEM photo, as was noticed originally. Hence, a chemical analysis of the components of the obtained polymers was also performed. On the basis of the obtained results, the presence of such elements as C, N, O, S was found (Figure 3).

Infrared (IR) spectra were recorded with an attenuated total reflection (ATR) mode in the 4000–400 cm^−1^ region. In the case of the FTIR analysis, the infrared spectrum consists mainly of two areas (Figure 4). The wide band, appearing at the value of the wave number $\bar{\nu }$ of approx. 3450 cm^−^^1^, corresponds to the stretching vibrations of the O–H group. The multiplet at $\bar{\nu }$ = 2964–2876 cm^−^^1^ corresponds to the stretching vibration of the C–H bonds. The intense signal at $\bar{\nu }$ = 1648 cm^−^^1^ is related to the vibration of C=C in the thiophene ring. Another relatively intense signal, occurring in the range $\bar{\nu }$ = 1384–1319 cm^−^^1^, is related to the vibrations originating from the tetrafluoroborate anion used in the polymer doping process to generate the appropriate charge. Another signal confirming the presence of this ion is the multiplet in the range of $\bar{\nu }$ = 1084–1034 cm^−^^1^. The sharp signal for the wavenumber $\bar{\nu }$ = 785 cm^−^^1^ corresponds to intra-plane deformation vibrations in the thiophene ring, and the multiplet in the range $\bar{\nu }$ = 674–623 cm^−^^1^, with much lower intensity, is related to out-of-plane C–H stretching vibrations. In the case of recording the spectra for poly (3-methylthiophene), the use of different electrodesorption solutions was compared. Bands characteristic of the case poly (3-methylthiophene) correspond to the bands described above (slight shift of the wavenumber values, $\bar{\nu }$). All samples were prepared as KBr pellets. Well-dried KBr does not show its own absorption in the range of $\bar{\nu }$ = 5000–400 cm^−^^1^, so all signals in the spectrum come only from the sample. By analyzing the obtained spectra, it can be concluded that the characteristic bandwidth broadening at the wavenumber value $\bar{\nu }$ of about 3450 cm^−^^1^ is associated with the formation of intermolecular hydrogen bonds between hydrogen and nitrogen (or sulfur) atoms present in the structure of polymers, and the functional groups of the analyzed drugs. The obtained IR spectra of synthesized polymer sorption fibers and the model molecule (template) fully confirmed the presence of functional groups that are characteristic of these coatings, polythiophene and poly (3-methylthiophene), respectively, and the appropriate drug template. In the analysis of each spectrum, apart from the characteristic vibration peaks from the monomers used and the stock solution, vibrations were also found from groups confirming the presence of the template molecule. The consequence of the presence of a given drug in the polymer matrix was the widening of the absorption band in the range of $\bar{\nu }$ = 3700–2800 cm^−^^1^. It was also observed that the intensity of the multiplet in the range of $\bar{\nu }$ = 1300–900 cm^−^^1^, characteristic of anions coming from the doping medium, also changed significantly. It can be concluded that the described characteristic vibration bands are responsible for the formation of interactions of the appropriate polymer sorption material with the pattern particles, and thus for the subsequent formation of imprints in the polymer matrix. The interpretation of the obtained infrared spectroscopy allowed for the recognition of structural bonds, including functional groups of samples of the tested sorption materials (Figure 4).

Additionally, in order to accurately determine the shape and morphology of the obtained polymer fibers, imaging using transmission electron microscopy (TEM) was performed. The conducted research revealed a granular, multi-phase structure of the tested materials. The grain size (d*_p_*), depending on the type of the polymer fiber, is 0.5–5.0 µm (Figure 5).

Due to the negligible solubility of polythiophene in the NMR solvents used, the research was carried out only for poly (3-methylthiophene). ^1^H NMR spectra made for this material are presented in Figure 6.

The pure 3-methylthiophene spectrum was used as a reference to identify PMeTh signals (data not shown). Due to the bonds between successive meras, multiplet signals in a chemical shift (δ) equal to about 7 ppm are replaced in PMeTh with a single singlet, while the resonance signal from the methyl group is observed in a similar chemical shift as in the case of pure 3-methylthiophene, i.e., about δ = 2.3 ppm. Two resonance signals potentially corresponding to the methyl group in the PMeTh polymer were observed in this region. In addition, a singlet potentially corresponding to the CH proton was observed in the chemical shift δ = 7.01 ppm. When integrated, the ratio of the above signals was in the range of 3:1–4:1, which may correspond to the ratio of protons of –CH_3_ to –CH groups.

In case of the SAXS measurements, the tests were performed after removing the sorption material from the working electrode [24]. As a result of using the low-angle X-ray scattering method, the porosity was determined for the tested SPME polymer sorption fibers without and with an imprinted particle. Figure 7 shows an example of a graph characterizing the size of the pores for poly(3-methylthiophene) fibers.

The pore size distribution was in the range between 40 and 120 nm (maximum value: SPME MIP-PTh—85 nm, SPME MIP-MePTh—90 nm). The maximum values in these areas were comparable with the results obtained for both PTh and PMeTh SPME fibers. The pore size distribution for the fibers with and without the imprinted drug molecule differed slightly. Hence, the differences in the efficiency of extraction are probably not the result of the morphology of the synthesized materials, but of the imprint effect.

Additionally, the chemical and mechanical stability of the synthesized SPME fibers was tested. For this purpose, various media were used: water, MeOH, THF, ACN, MeOH/H_2_O mixture (50:50, *v/v*), 2-propanol, MeOH/6.5 mM CH_3_COOH (75:25, *v/v*) and MeOH/25% NH_3_ (80:20, *v/v*) mixture to evaluate these properties. There were no meaningful differences in the structure of a given SPME sorption fiber before and after the exposure to a given medium.

### 3.3. Optimization of MIP-SPME Extraction through Experimental Design (DOE)

A design of the experiment (DOE) method was applied to optimize the electrochemical polymerization parameters. In this study, the CCD was successfully applied to evaluate the relevance of the three controlled parameters for the MIP-SPME extraction and to identify possible interactions between them. It is described by the following model (Equation (1)):Y = b_0_ + b_1×1_ + b_2_X_2_ + b_3_X_3_ + (b_12_X_1_X_2_ + b_13_X_1_X_3_ + b_23_X_2_X_3_) + b_11_X_1_^2^ + b_22_X_2_^2^ + b_33_X_3_^2^(1)

Optimized input variables (X_1_, X_2_ and X_3_ in the equation) were adsorption time, desorption time and the volume of the desorption medium. For optimizations, the response (Y) was the antibiotic drug peak area at the extracted ion chromatogram (EIC) for the precursor ion. Moreover, b_i_ coefficients in the equation represent the linear terms, b_ij_ the interactions and b_ii_ the coefficients of quadratic terms.

The methodology with the desorption time (5–15 min), the pH of the desorption solvent (3–10) and the volume of the desorption solvent (30–100 μL) as the three variables requires a significantly smaller number of experiments than the traditional “one variable at a time” methodology. The CCD was executed requiring 15 experiments. The surface mapping was an effective way to graphically localize the optimal conditions, allowing for the highest EIC peak area for each target compound. 3D contour plots were prepared for this purpose (Figure 8).

Owing to the graphs, which show interactions and dependencies between two variables and extraction efficiency, it was possible to indicate the best fit of conditions depending on the values of X_1_–X_3_. The greater the desoption time value (X_1_), the better the system fit and the larger the surface area. The X_2_ and X_3_ values, the pH of the desorption solvent and the volume of the desorpion solvent do not significantly affect the fit. The fit depends mainly on the value of X_1_. Initially, the larger the value of X_1_, the better the fit; however, once the optimal value for the best fit is reached, a further increase in X_1_ results in a deterioration of the result. The conclusion is that, with the same parameters, similar selectivity can be obtained. It has to be pointed out that for all the target compounds, the results were very similar, indicating the same behavior during the MIP-SPME extraction.

Finally, MIP-SPME coatings were preconditioned in the mixture of MeOH and water (80:20, *v/v*) for 15 min, followed by water with 0.1% formic acid (for 10 min). The adsorption step of the target compounds was carried out in the direct immersion mode in 50 μL of human plasma during 10 min without agitation (at 37 °C ± 0.2). The desorption step was performed with the use of 65 μL ammonium acetate in 70:30 MeOH:H_2_O (for 10 min). The evaporated samples were dissolved in the 180 μL of mobile phase and analyzed with the use of liquid chromatography and mass spectrometry.

### 3.4. Proposed Mechanism of Drug Sorption on MIP-SPME Fibers

The Figure 9a–c below presents the proposed mechanism of interaction between a given drug and the SPME polymer sorption fiber. Figure 9a shows possible interactions between the cefotaxime molecule and the polythiophene film. This type of fiber has a sulfur atom in its structure, which is an element with high electronegativity, along with free electron pairs, just like the oxygen atom from the O=C-carbonyl group derived from the drug molecule. In this case, there is no effect. Presumably, the only interactions that occur here are the donor–acceptor interaction between the hydrogen atom of the H–N–group and the H_2_C-moiety of the drug molecule and the sulfur atom of the thiophene ring (in the case of polythiophene). Additionally, very weak hydrogen bonds (due to the spherical interactions in the thiophene ring) presumably can be formed between the hydrogen atoms of the polymer chain and the oxygen atom of the O=C-carbonyl group of the drug molecule. In the case of metronidazole (Figure 9b) one can expect the donor–acceptor interaction between the oxygen atom from the hydroxyl group of the HO–drug molecule and the sulfur atom from the thiophene ring (in the case of polythiophene). Moreover, this also applies to the interaction between the nitrogen atom of the nitro group and the hydrocarbon drug molecule, along with the sulfur atom from the thiophene ring (in the case of polythiophene). In the case of a surface covered with poly (3-methylthiophene) and amoxicillin (Figure 9c), the formation of hydrogen bonds between the hydrogen atom of the H-C-polymer group and the oxygen atom of the O=C-carbonyl group of the drug molecule and the amino group HN-entails a possible drug molecule and the sulfur atom of the methylthiophene ring (in the case of poly (3-methylthiophene)).

On the basis of the obtained results, it was observed that depending on the type of the polymer (PTh, PMeTh), the sorption properties of the SPME coatings are different. Moreover, the thickness of the MIP-SPME layer is very important for the printed patterns (in case of drugs) and for the isolation and labeling of the analytes. It is likely that the thicker this layer is, the greater the number of molecular cavities imprinted in it. Thus, the ability of the polymer to interact with the analyte is greater. However, the problem of removing all the printed templates remains, because their being washed out incompletely would reduce the sorption efficiency during the determination of the analytes.

### 3.5. Application of SPME HPLC-MS/MS to Human Plasma Samples

The MS spectra for the human plasma samples are shown in Figure 10. Moreover, we studied the fragmentation of the [M + H]^+^ ions of the target compounds and their metabolites with the use of ESI-MS/MS measurements. Based on the obtained results, we can notice that the MS spectra are clear, which demonstrates that the MIP-SPME approach was efficient and selective.

In the real samples from patients treated with metronidazole, the presence of one metabolite of the parent drug—hydroxymetronidazole, *m/z* = 188 (for MET-OH)—was observed. For amoxicillin and cefotaxime, four metabolies were identified at *m/z* = 366 and 384 (for AMD and AMA, respectively) and at *m/z* = 396 (for CEF-dAce and CEF-dAceLac). The acquired MS/MS spectra proved that the precursor ion for MET at *m/z* = 172 was mainly fragmented to *m/z* = 128 (–C_2_H_4_O) and for MET–OH at *m/z* = 188 to *m/z* = 144 (–C_2_H_2_NO) and *m/z* = 126 (–C_2_H_4_O). For CEF and CEF-DAC-LAC, the precursor ion was at *m/z* = 456 and *m/z* = 396, respectively. The CEF fragmentation (MS/MS) created mainly two product ions identified at *m/z* = 342 and *m/z* = 396 (major product) and CEF-DAC-LAC fragmented to *m/z* = 282 and *m/z* = 336.

The influence of different parameters, which were important from the point of view of the physiology of the human body, on the sorption efficiency of the selected antibiotic was investigated. As we know, the most important organic blood components are albumin (the protein fraction representing about 60%). Thus, it was necessary to study their interaction on the extraction efficiency of the analyzed drugs. A slight influence of albumin (c_physiological_ = 30–50 mg/mL) on the sorption process in the case of the tested antibiotics was noticeable. On the basis of the obtained results, it was also possible to eliminate the influence of cations and anions on the sorption properties of polymer fibers. The need to investigate their interaction was due to the fact that, in plasma, which accounted for 55% *vol.* in the whole blood, apart from morphotic elements (erythrocytes, leukocytes, platelets), inorganic components were suspended, mainly sodium and chloride ions. Depending on the use of an appropriate dilution of sodium chloride solution (in the concentration range of 0.225–0.9%) or a magnesium-calcium buffer, the sorption efficiency was maintained at the same level.

The total amount of extracted drug (M_cał_) per MIP-SPME fiber was calculated from the equation: $M_{\mathrm{cał}}=c_{\mathrm{des}}\cdot V_{\mathrm{des}}$, where c_des_ is the drug concentration in the desorbed solution [μg/mL] and V_des_ is the volume of the desorbed solution [mL]. By comparing the amounts of the adsorbed drug at a concentration of 1 μg/mL, in the human plasma samples, these values are: MIP-PMeTh (AMOX)—1.26 μg, MIP-PTh (CEF)—1.24 μg, MIP-PTh (MET)—1.19 μg.

Moreover, the reproducibility of the MIP-SPME coatings was investigated with the use of the same concentration of the drug solution (for AMOX, CEF and MET, 1 μg/mL) and six fibers synthesized under the same conditions. The extraction efficiency revealed the relative standard deviation (RSD) values, app. 3.5%, confirming that the data are highly reproducible. Based on Table 3, it can be noticed that RDS is in the range of 0.12–3.46%. By analyzing the amounts of adsorbed AMOX on individual MIP-SPME fibers and comparing the linear correlation coefficients, it can be concluded that the most effective extraction is obtained for a fiber with a PMeTh coating, and the least effective is for a PTh fiber. The reverse situation is in the case of CEF and MET. The reason for this is the interaction between the drug and a given SPME fiber coverage. It can be concluded that MIP-SPME fibers with porous material have the ability to retain characteristic molecules in their pores more strongly. This means that only selected substances from the tested medium can bind to the carrier, while analytes with a different structure are not captured. Thus, the effect of matching the particle size and pores of the sorbent will play a key role.

## 4. Conclusions

The presented research results seem to confirm the sorption properties of electroconductive polymeric materials with an imprinted particle (MIP-SPME). This creates new possibilities in the field of the selective extraction of drugs and their potential metabolites. In addition, the use of polymer sorption fibers significantly improves the isolation of biologically active compounds remaining in low concentrations in the initial matrices. The high selectivity of SPME sorbents with an imprinted particle makes it possible to use them for the isolation from real biological samples (plasma/whole blood). This approach shortens the sample preparation time and minimizes the amount (volume) of the biological material necessary for analytical purposes.

Moreover, recent trends set by the so-called “green chemistry principles” in extraction methods focus primarily on finding solutions that reduce solvent consumption and improve the efficiency and selectivity of the whole process. One such solution is to apply the described analytical procedure with the use of solid-phase microextraction. Both processes were optimized to define the impact of the extraction parameters of the SPME approach and conditions of the electropolymerization step on the extraction efficiency of target compounds. Then, the chemical profiles of the analyzed human plasma samples from the patients were determined with the use of LC-MS/MS.

## Figures and Tables

**Figure 1 materials-14-04886-f001:**
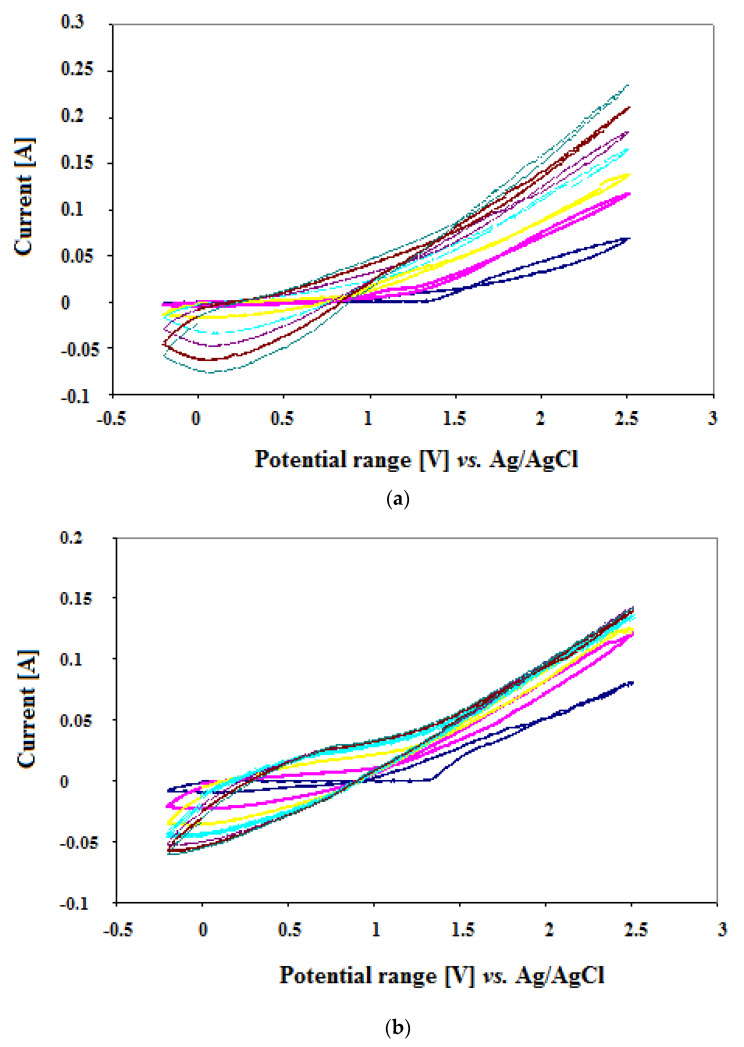
Voltamperograms obtained during the electropolymerization process of thiophene, with (**a**) and without (**b**) imprinted drug molecule (CEF). (**a**) demonstrates cyclic scans of electropolymerization of thiophene in the presence of CEF and (**b**) without CEF. Colors mean seven-cycle scans during polymerization.

**Figure 2 materials-14-04886-f002:**
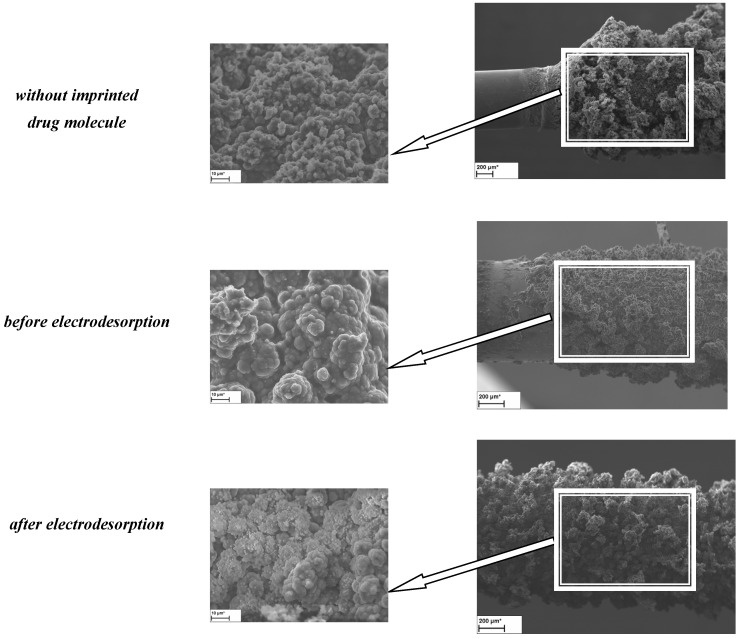
Microscopic photos of fibers with polythiophene coating during various stages of electropolymerization (for MET).

**Figure 3 materials-14-04886-f003:**
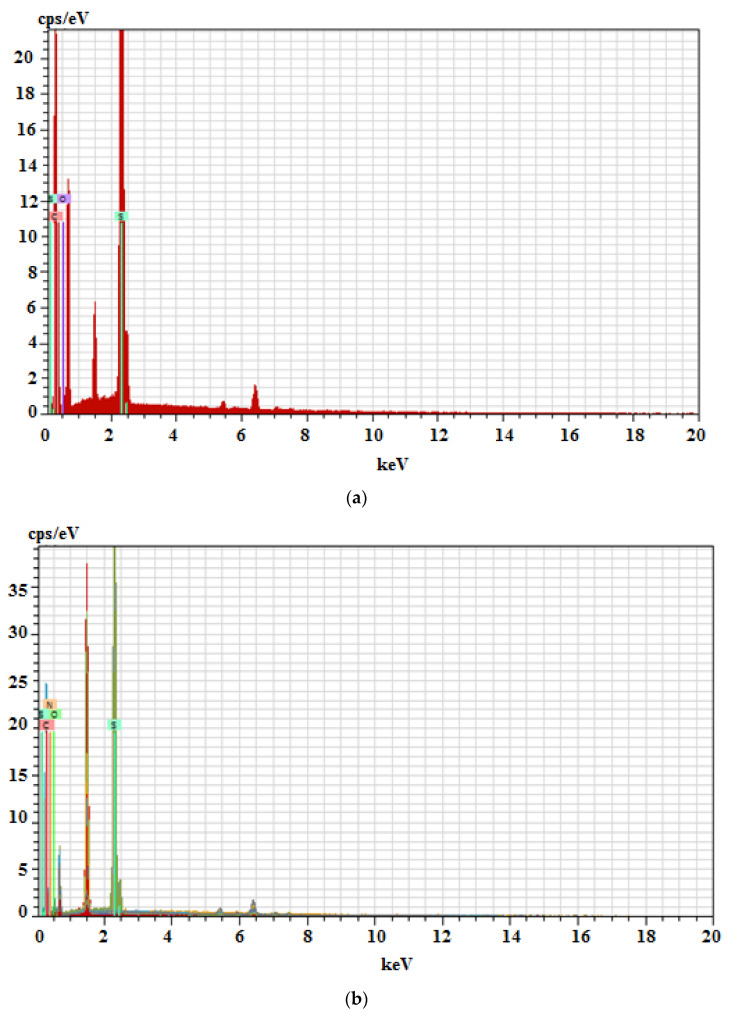
EDX chemical analysis of the poly (3-methylthiophene) surface without (**a**) and with imprinted drug molecule (for AMOX) (**b**).

**Figure 4 materials-14-04886-f004:**
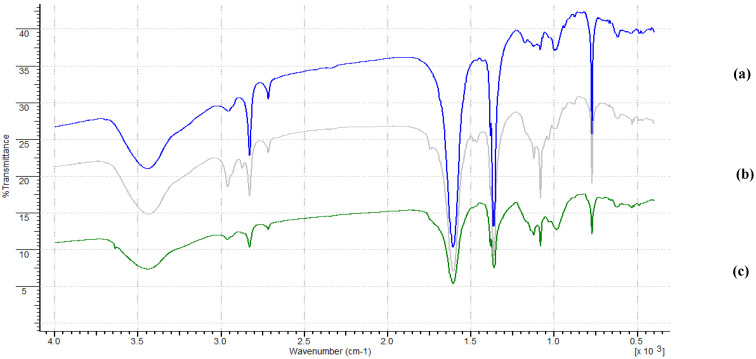
FTIR spectrum of the polythiophene SPME fiber before electrodesorption (**a**), after electrodesorption (**b**) and without imprinted drug molecule (**c**) (for CEF) (KBr pelleting technique).

**Figure 5 materials-14-04886-f005:**
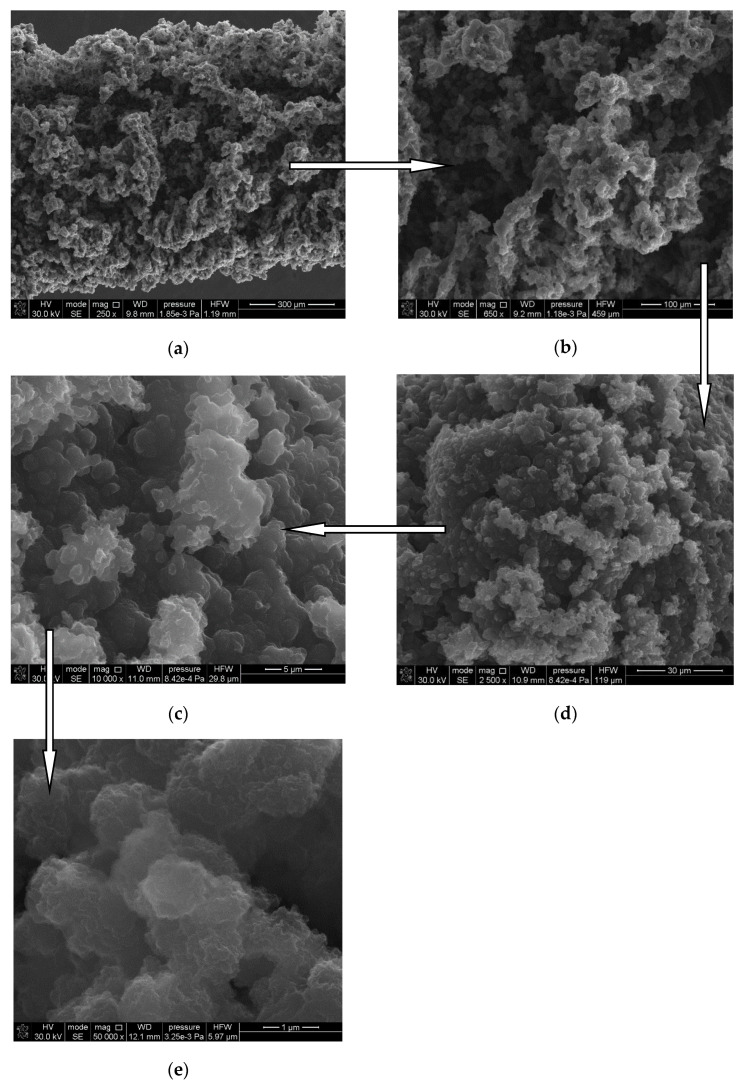
Structure and morphology of the surface of the sorption polythiophene MIP-SPME fiber under a transmission electron microscope (for MET).

**Figure 6 materials-14-04886-f006:**
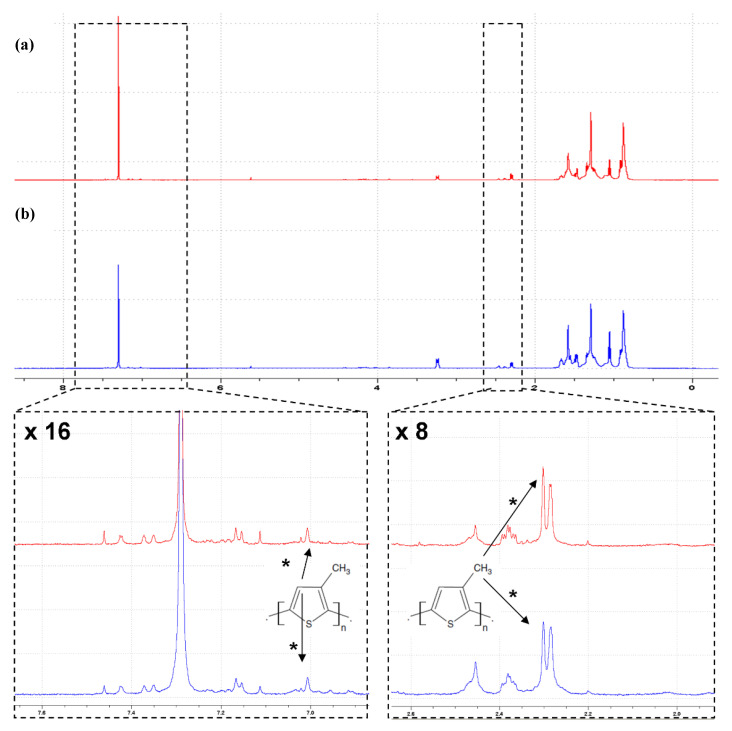
^1^H NMR spectra for poly (3-methylthiophene) fibers, with (**a**) and without (**b**) imprinted drug molecule (for AMOX).

**Figure 7 materials-14-04886-f007:**
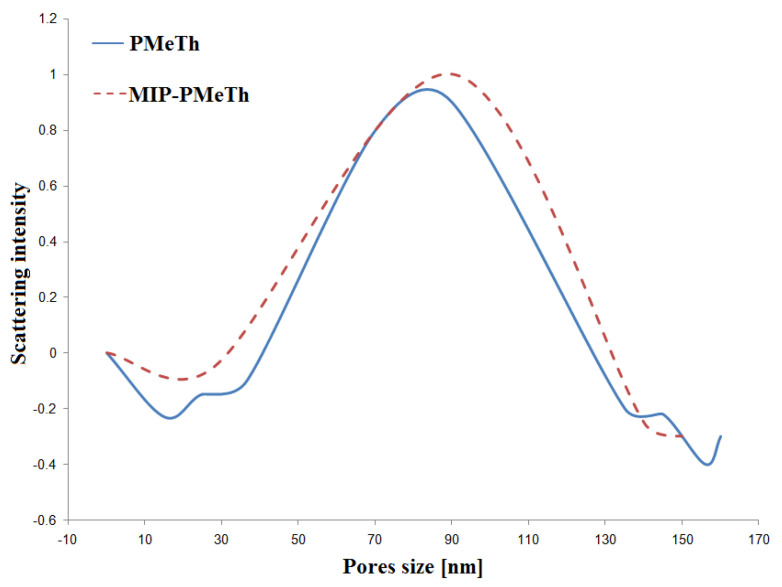
SPME-PMeTh fiber porosity measurement with SAXS (for AMOX).

**Figure 8 materials-14-04886-f008:**
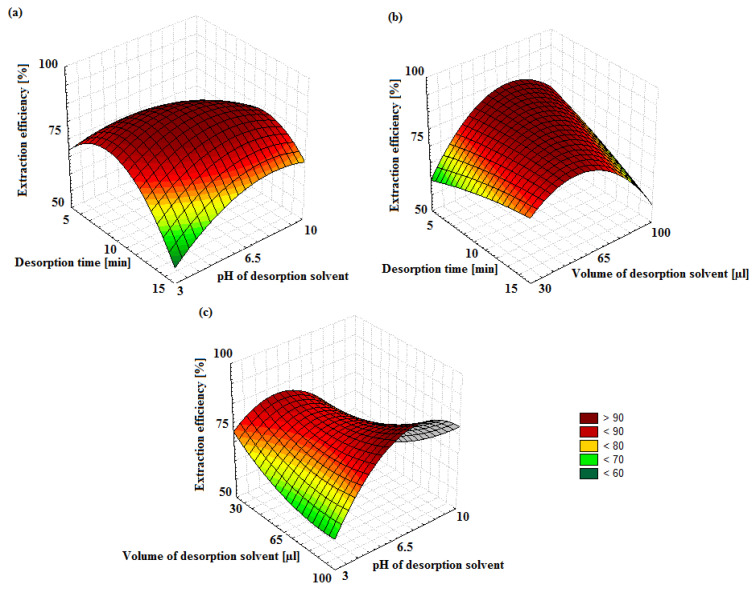
Response surface methodology for the extraction recovery of AMOX in case of optimized parameters such as the desorption time (X_1_), the pH of the desorption solvent (X_2_) and the volume of the desorption solvent (X_3_). (**a**) X1 vs. X2, (**b**) X1 vs. X3, (**c**) X3 vs. X2.

**Figure 9 materials-14-04886-f009:**
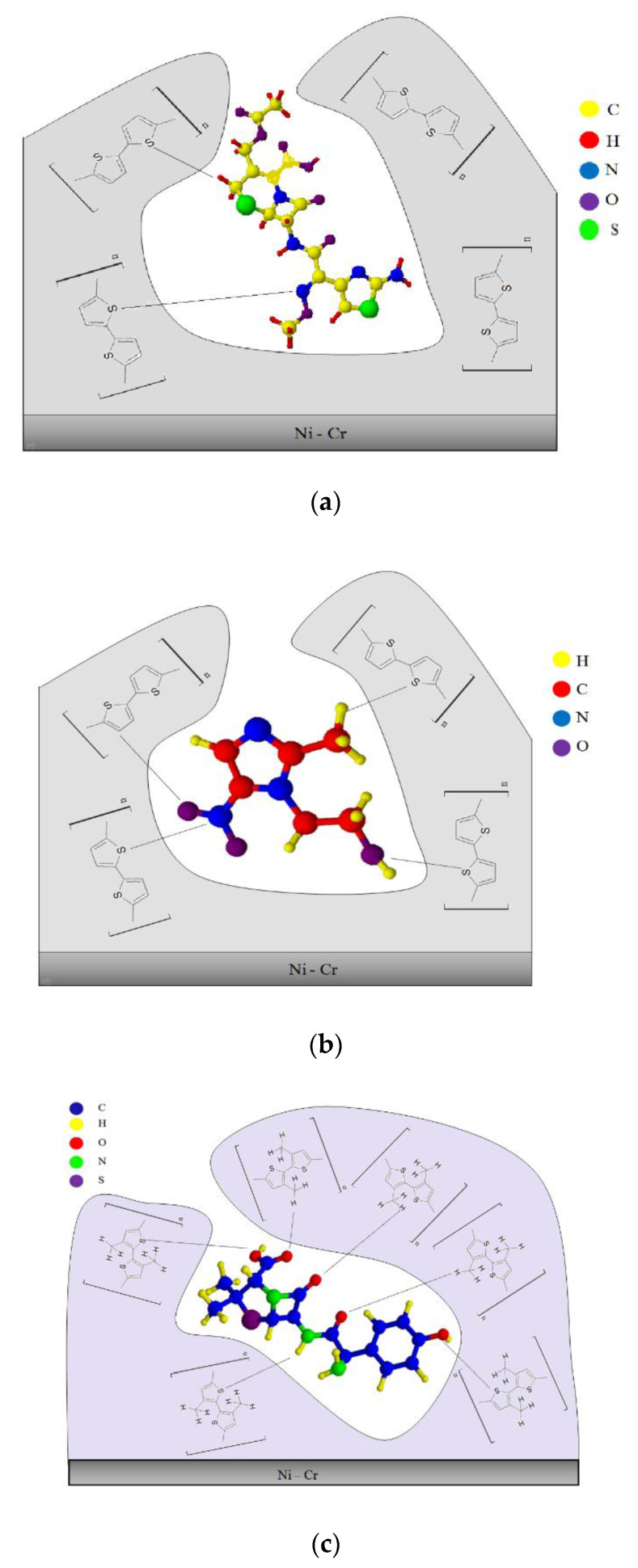
Proposed mechanism of the sorption of cefotaxime (**a**), metronidazole (**b**) and amoxicillin (**c**) molecules on the surface of PTh-SPME fiber and PMeTh-SPME fiber.

**Figure 10 materials-14-04886-f010:**
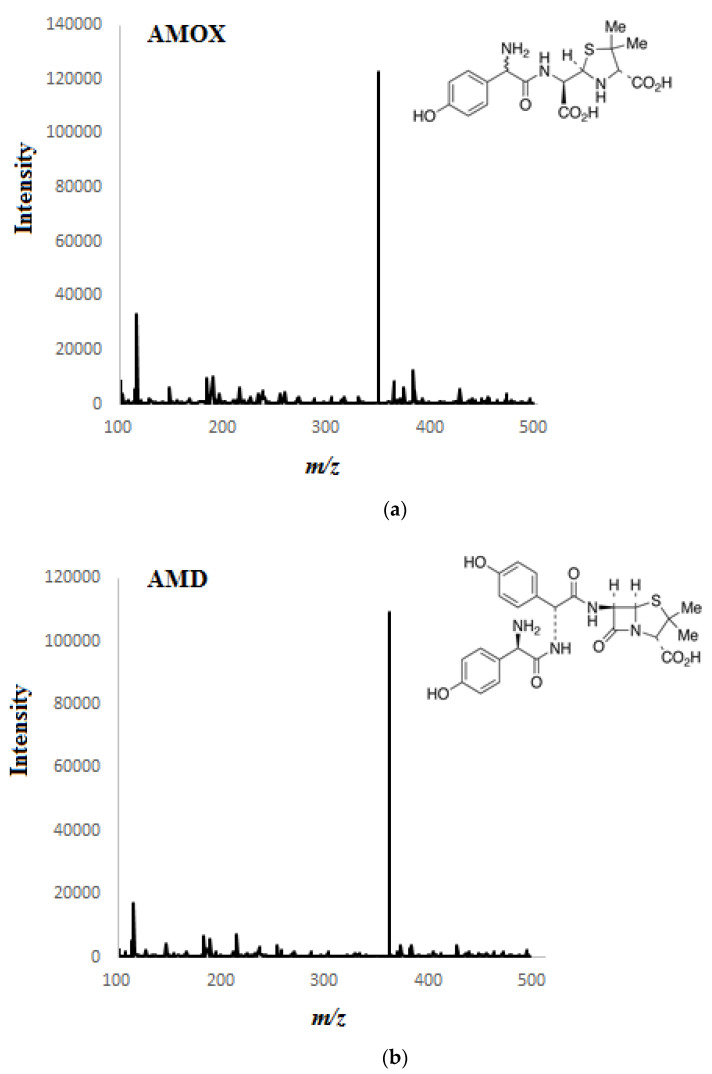
MS spectra for the human plasma samples analyzed after the oral administration of AMOX (**a**), CEF (**d**) and MET (**g**), showing drugs and their metabolites (AMD (**b**), AMA (**c**), CEF-dAceLac (**e**), CEF-dAce (**f**), MET-OH (**h**)).

**Table 1 materials-14-04886-t001:** Parameters of the electrochemical synthesis of polythiophene and poly(3-methylthiophene) coatings.

Parameter	Monomer	
	*Thiophene*	*3-methylthiophene*
Potential range *[V* vs. *Ag/Ag^+^]*	−0.2 ÷ 2.5	−0.2 ÷ 3.0
Polymerization rate *[mV/s]*	50	50
System balancing time *[s]*	0	0
Number of repetitions (scans)	7	3
Monomer concentration *[mol/dm^3^]*	0.40	0.10
Base electrolyte concentration *[mol/L]*	0.1	0.1
The length of the fiber coverage *[mm]*	15.0	15.0
Thickness of the fiber coverage *[μm]*	285–295	110–160
Organic carbon content *[%]*	45.9	57.2
Concentration of the functional monomer *[mol/L]*	2.5/PThCEF/ 3.0/PThMET/	1.5/PMeThAMOX/
Template concentration *[μg/mL]*	0.4/CEF/ 0.35/MET/	0.1/AMOX/
Medium used for electrodesorption	0.05 M NaOH/MeOH /PThCEF, PThCEF/	0.2 M Na_2_HPO_4_/EtOH/PMeThAMOX/
Potential range, MIP *[V* vs. *Ag/Ag^+^]*	0–1.2	0–1.2
**Polymer (product)**	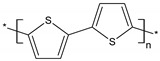	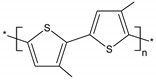

**Table 2 materials-14-04886-t002:** Mass spectrometry parameters applied for the determination and identification of relevant antibiotics and their metabolites in human plasma.

Compound	Molecular ion, *m/z*	*Q1*, *m/z*	*Q3*, *m/z*	CE (eV); for *Q1*, *Q3*	DGT (°C)	F	CV (V)
**AMOX** β-lactams	366	114	349	22, 36	320	150	3500
**AMA** metabolite	384	189	323	27, 24	320	150	4000
**AMD** metabolite	366	134	337	29, 35	320	150	4000
**CEF** β-lactams	456	396	342	29, 30	290	110	4500
**CEF-DAC LAC** metabolite	396	336	282	31, 28	290	110	4500
**MET** nitroimidazoles	172	128	-	25	320	150	3500
**MET-OH** metabolite	188	126	144	29, 31	320	150	3500

*Q1*—Quantifier ion; *Q3*—Qualifier ion; CE—Collision energy; DGT—Drying gas temperature; F—Fragmentor; CV—Capillary voltage.

**Table 3 materials-14-04886-t003:** Extraction efficiency for AMOX, CEF and MET in case of PTh and PMeTh SPME Coatings (*n* = 6).

Compound	MIP-SPME Coating	Slope	Intercept	R^2^	RSD [%]
*AMOX*	PTh	0.0219	0.2884	0.9193	0.12–2.46
	PMeTh	0.0704	0.1783	0.9994	0.26–1.49
*CEF*	PTh	0.0814	0.1934	0.9991	0.37–1.76
	PMeTh	0.0352	0.2918	0.8465	0.86–3.46
*MET*	PTh	0.0946	0.8375	0.9997	0.17–2.09
	PMeTh	0.0517	0.3946	0.8493	0.65–3.05

## Data Availability

Not applicable.

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
