# Peer review of "Molecularly Imprinted Polymers as Solid-Phase Microextraction Fibers for the Isolation of Selected Antibiotics from Human Plasma"

_materials, 2021, doi:10.3390/ma14174886_

Round 1

Reviewer 1 Report

The submitted manuscript describes and discusses the results of an original research project carried out to synthesize and characterize two novel molecularly imprinted polymers (MIP), consisting of a polythiophene and a poly(3-methylthiophene) coated solid phase microextraction fiber (MIP-SPME). Both novel materials have been produced by a direct electrochemical polymerization process, whose optimization has been performed by a design of experiment (DOE) approach. Also described and discussed is the successful application of the synthesized MIP-SPME fibers to the selective extraction of three target antibiotic drugs and their main metabolites from human plasma samples, collected from patients of an intensive care unit.

The manuscript describes and discusses logically designed experiments and presents results that are expected to be of large interest for the scientific community. Nevertheless, the manuscript needs a minor revision. None of the papers listed in the References section have been cited in the text, also containing typographical and careless errors, most of which are specified below.

Line 96: source of the water employed to carry out all experiments has already been reported in lines 90-92.

Table 1 and throughout the text: special characters, such as µ in µg and µL are missed. SI units and their symbols should be used (liter instead than dm3; L instead than l).

Section 2.5.3: The IR instrument listed in this section is not that mentioned in the “Results and Discussion” section (see line 314).

Legend of Figure 6: a clear description of the different panels displayed in the figure should be included.

Line 432: the volume ratio of MeOH/CH3COOH and MeOH/NH3, as well as the concentration of acetic acid and ammonia, are not reported.

Author Response

The submitted manuscript describes and discusses the results of an original research project carried out to synthesize and characterize two novel molecularly imprinted polymers (MIP), consisting of a polythiophene and a poly(3-methylthiophene) coated solid phase microextraction fiber (MIP-SPME). Both novel materials have been produced by a direct electrochemical polymerization process, whose optimization has been performed by a design of experiment (DOE) approach. Also described and discussed is the successful application of the synthesized MIP-SPME fibers to the selective extraction of three target antibiotic drugs and their main metabolites from human plasma samples, collected from patients of an intensive care unit.

The manuscript describes and discusses logically designed experiments and presents results that are expected to be of large interest for the scientific community. Nevertheless, the manuscript needs a minor revision. None of the papers listed in the References section have been cited in the text, also containing typographical and careless errors, most of which are specified below.

RE: Indeed, we agree with the Referee. The revised manuscript has been checked and corrected by English Native Speaker.

Line 96: source of the water employed to carry out all experiments has already been reported in lines 90-92.

RE: Thank you for your remarks. It was modified.

Table 1 and throughout the text: special characters, such as µ in µg and µL are missed. SI units and their symbols should be used (liter instead than dm3; L instead than l).

RE: Thank you for your remarks. It was modified.

Section 2.5.3: The IR instrument listed in this section is not that mentioned in the “Results and Discussion” section (see line 314).

RE: Thank you for your remarks. It was modified.

Legend of Figure 6: a clear description of the different panels displayed in the figure should be included.

RE: Thank you for your remarks. It was modified.

Line 432: the volume ratio of MeOH/CH3COOH and MeOH/NH3, as well as the concentration of acetic acid and ammonia, are not reported.

RE: Thank you for your remarks. It was modified.

Thank you very much for your critical review. It was very useful in the correction of our manuscript. Identification of weak points throughout the text has helped us to increase the value of our paper. All comments and changes suggested by Reviewers have been incorporated into the manuscript. Once again, thank you very much for your help.

Reviewer 2 Report

In this study, the authors used the electrochemical polymerization method to prepare novel MIP-coated SPME fibers for removing antibiotics from human plasma samples. It is an interesting study with a novel approach. The paper in the whole is well designed and results sound.

              My comments and doubts are:

  1. In the electropolymerization process, why the authors used a different monomer (3-methylthiophene) for amoxicillin. Why they didn´t use thiophene with this antibiotic?
  2. In line 235, the authors claim “Its absence indicates that during the electropolymerization process the target drug template was imprinted in the MIP layer in its unchanged form”. In my opinion, for the reader, it will be more interesting see the cyclic voltammograms taken during the electropolymerization of thiophene with and without antibiotic (Figure 1).
  3. In Figure 2, what was the antibiotic used? Please add the name of antibiotic.
  4. In the caption of Figure 3, the authors showed the EDX spectrum for 3-methylthiophene and for cefotaxime (CEF). Is there not a mistake here? Shouldn´t be amoxicillin? Even so, since the authors previously showed the SEM images of thiophene, in my opinion, the authors should show the EDX spectrum for the thiophene instead for 3-methylthiophene . Thus, the authors will have access to the full characterization of a system (for example: thiophene+CEF) and may avoid misunderstandings.
  5. In the caption of Figure 4, please insert the name of model molecule.
  6. In the caption of Figure 5, please add more details such as: pure or MIP- SPME fiber? Monomer used? Model molecule used?
  7. In the manuscript, the authors didn´t discuss the Figure 8. The authors stated, in line 479, “These plots are simple graphic representations of the regression equation for the optimization of MIP-SPME parameters, which presents dependencies between each two independent variables and the EIC peak area.” In the manuscript, the authors can describe the graphs obtained for amoxicillin.
  8. After reading the manuscript, I was wondering if the authors tested the selectivity and reproducibility of these MIP-SPME fibers. I mean, the authors tested the selectivity of these fibers in the presence of interfering molecules? Interfering molecules can be present in biological sample (e.g. proteins) and may interfere with the determination of these antibiotics.
  9. Regarding extraction efficiency of antibiotics from human samples: what was the amount of extracted antibiotic with the use of these SPME fibers? What was the MIP-SMPE fibers that provided better sorption efficiencies?

Author Response

In this study, the authors used the electrochemical polymerization method to prepare novel MIP-coated SPME fibers for removing antibiotics from human plasma samples. It is an interesting study with a novel approach. The paper in the whole is well designed and results sound.

              My comments and doubts are:

  1. In the electropolymerization process, why the authors used a different monomer (3-methylthiophene) for amoxicillin. Why they didn´t use thiophene with this antibiotic?

RE: Thank you for your remarks. It was added to the manuscript.

  1. In line 235, the authors claim “Its absence indicates that during the electropolymerization process the target drug template was imprinted in the MIP layer in its unchanged form”. In my opinion, for the reader, it will be more interesting see the cyclic voltammograms taken during the electropolymerization of thiophene with and without antibiotic (Figure 1).

RE: Thank you for your remarks. It was added.

  1. In Figure 2, what was the antibiotic used? Please add the name of antibiotic.

RE: It was modified.

  1. In the caption of Figure 3, the authors showed the EDX spectrum for 3-methylthiophene and for cefotaxime (CEF). Is there not a mistake here? Shouldn´t be amoxicillin? Even so, since the authors previously showed the SEM images of thiophene, in my opinion, the authors should show the EDX spectrum for the thiophene instead for 3-methylthiophene . Thus, the authors will have access to the full characterization of a system (for example: thiophene+CEF) and may avoid misunderstandings.

RE: It was modified.

  1. In the caption of Figure 4, please insert the name of model molecule.

RE: It was modified.

  1. In the caption of Figure 5, please add more details such as: pure or MIP- SPME fiber? Monomer used? Model molecule used?

RE: It was modified.

  1. In the manuscript, the authors didn´t discuss the Figure 8. The authors stated, in line 479, “These plots are simple graphic representations of the regression equation for the optimization of MIP-SPME parameters, which presents dependencies between each two independent variables and the EIC peak area.” In the manuscript, the authors can describe the graphs obtained for amoxicillin.

RE: It was modified.

  1. After reading the manuscript, I was wondering if the authors tested the selectivity and reproducibility of these MIP-SPME fibers. I mean, the authors tested the selectivity of these fibers in the presence of interfering molecules? Interfering molecules can be present in biological sample (e.g. proteins) and may interfere with the determination of these antibiotics.

RE: Thank you for your remarks. It was added.

  1. Regarding extraction efficiency of antibiotics from human samples: what was the amount of extracted antibiotic with the use of these SPME fibers? What was the MIP-SMPE fibers that provided better sorption efficiencies?

RE: Thank you for your remarks. It was added.

Thank you very much for your critical review. It was very useful in the correction of our manuscript. Identification of weak points throughout the text has helped us to increase the value of our paper. All comments and changes suggested by Reviewers have been incorporated into the manuscript. Once again, thank you very much for your help.

Reviewer 3 Report

file Attached

Author Response

The manuscript titled, “Molecularly imprinted polymers coated solid phase microextraction fibers for the isolation of antibiotics from human plasma patients in the intensive care unit” by Szultka-MÅ‚yÅ„ska et. al reported the synthesis of novel molecularly imprinted polymer-coated
polythiophene and poly(3-methylthiophene) solid phase microextraction fibres, their
characterization, and applications towards detection of antibiotics and their metabolites. The
authors claimed that this is first study using molecularly imprinted polymers coated solid phase microextraction fibres for recognition of antibiotics and their metabolites in real samples from patients. Therefore, the manuscript addresses the potential interest of the study. However, the data set provided in this manuscript is not sufficient to support the conclusion and thus it seems premature to proceed with the manuscript based on the current results. Therefore, I would recommend rejection.

RE: Indeed, we agree with the Referee. It was modified.

Figure 2 displaying scanning electron microscopy (SEM) images of said polymer fibres are not displaying any fibrous morphologies as mentioned in manuscript. SEM samples were deposited on solid Ni-Cr surface. But surprisingly it wasn’t detected in EDX spectra.

RE: We did not mentioned in the manuscript about ‘fibrous morphologies’. Just SPME/MIP-SPME fibers in the meaning of carrier/coating. In case of EDX the experiments were performed after removing the sorption material (PTh, PMeTh) from the working electrode (SS, Ni-Cr).

Figure 5, transmission electron microscopy (TEM) images are not displaying any fibrous
morphologies as mentioned in manuscript.

RE: We did not mentioned in the manuscript about ‘fibrous morphologies’. Just SPME/MIP-SPME fibers in the meaning of carrier/coating.

Figure 6 showing comparison of two NMRs need detailed explanations. The authors should
submit an NMR spectrum using DMSO-d6 solvent.

RE: In the case of the polymer samples, it was necessary to dissolve them with a deuterated solvent. Each sample was suspended in 1 ml of CDCl3 and placed in an ultrasonic bath. The polymer coatings were shredded under the influence of ultrasound (30 min). Due to the negligible solubility of polythiophene in the NMR solvents used, the research was carried out only for poly (3-methylthiophene) in CDCl3.

LC-MS technique is well established for detection of small molecules and antibiotics from
blood/serum samples. Some of recently published literature: (a) Neugebauer, Sophie, Christina Wichmann, Sibylle Bremer-Streck, Stefan Hagel, and Michael Kiehntopf. "Simultaneous quantification of nine antimicrobials by LC-MS/MS for therapeutic drug monitoring in critically ill patients." Therapeutic drug monitoring 2019, 41, 29. (b) Rehm, S. and Rentsch, K.M., “LC-MS/MS method for nine different antibiotics.” Clinica Chimica Acta, 2020, 511, 360. What is the advantage of using Molecularly imprinted polymers coated solid phase microextraction fibers?

RE: For the analysis of antibiotic drugs in pharmaceutical and clinical samples, sample preparation is a critical step before instrumental analysis. The sample preparation procedure for the determination of antibiotic drugs in pharmaceutical and clinical samples typically involves several steps such as extraction and/or dilution of the sample, preconcentration of the analytes, and removal of interfering substances. Liquid-liquid extraction and solid phase extraction are widely used for antibiotics analysis. Solid phase microextraction, as miniaturized sample preparation technique, is also applied for antibiotics analysis. Conducting polymers (PTh, PMeTh) as advanced materials have been known to possess great application potential in separation science owing to their versatile properties such as hydrophobicity, large π-conjugated structure, hydrogen bonding, ion exchange property, electroactivity, and so on. Among these polymers, PTh and its derivatives have been studied more due to their good environmental stabilities, facile synthesis, extraction capability toward polar compounds, and relatively low cost. The application of the MIP-SPME procedure at the preparation stage provided satisfying extraction efficiency. Moreover, the good sensitivity and precision and the high accuracy of the assay of selected antibiotic drugs make the MIP-SPME/LC–MS method a useful tool in clinical laboratories for therapeutic drug monitoring of antibiotics as well as in forensic laboratories for their determination at therapeutic and higher levels in human fluids. Among methods available in the literature, proposed method provided better accuracy and precision than solid phase extraction or protein precipitation. MIP-SPME as a new sample preparation technique is suitable for the determination of analytes in complex matrices, and requires only small sample volumes. Reduction of sample volume was also possible, because the MS detection was applied. Applied sample preparation technique is simply to perform, limiting the volumes of organic solvents and biological samples as well.

There is lack of important reference citations related to the study.

RE: Indeed, we agree with the Referee. It was modified.

Thank you very much for your critical review. It was very useful in the correction of our manuscript. Identification of weak points throughout the text has helped us to increase the value of our paper. All comments and changes suggested by Reviewers have been incorporated into the manuscript. Once again, thank you very much for your help.

Reviewer 4 Report

In this work the polymer fibers coated with drug molecules were synthesized for solid phase microextraction (SPME) purpose. Specifically, two different types of electroconductive polymer coatings with and without drugs attached were prepared by electrochemical based polymerization and characterized by different physical, chemical and morphological techniques. The drug imprinted polymer coated on metal supports were tested in human plasma and other samples for the extraction study. The results show that polymer coatings were successfully synthesized. The extraction results show that the drugs were adsorbed to the polymer surface and then desorbed for analytical purpose, which demonstrates that this electroconductive polymer material has potential in biological molecule extraction applications. The introduction, experimental design, data analysis and results discussion are complete. I would suggest the publication of this work. My comments are listed below:

  1. Is there any data and any control experiment support that the drug adsorption is “highly selective” to these particular polymers? In other words, do the drugs also adsorb to other polymers?
  2. There should be a discussion about what properties of the polythiophene and poly(3-methylthiophene) layer could affect the drug sorption behavior. i.e., thickness, smoothness, net charge.
  3. Some minor points:

Line 363, it should be “0.5 – 5.0” not “0.5 ÷ 5.0”. Same for line 424;

A high-resolution picture is suggested for Fig 3;

In the Results and Discussion section, there are some descriptions regarding the techniques (i.e., the operation principle and scan parameter of SEM and FTIR.) that can be moved to the supplemental file or the previous Materials and Methods section.

Author Response

In this work the polymer fibers coated with drug molecules were synthesized for solid phase microextraction (SPME) purpose. Specifically, two different types of electroconductive polymer coatings with and without drugs attached were prepared by electrochemical based polymerization and characterized by different physical, chemical and morphological techniques. The drug imprinted polymer coated on metal supports were tested in human plasma and other samples for the extraction study. The results show that polymer coatings were successfully synthesized. The extraction results show that the drugs were adsorbed to the polymer surface and then desorbed for analytical purpose, which demonstrates that this electroconductive polymer material has potential in biological molecule extraction applications. The introduction, experimental design, data analysis and results discussion are complete. I would suggest the publication of this work. My comments are listed below:

  1. Is there any data and any control experiment support that the drug adsorption is “highly selective” to these particular polymers? In other words, do the drugs also adsorb to other polymers?

RE: Thank you for your remarks. It was added.

  1. There should be a discussion about what properties of the polythiophene and poly(3-methylthiophene) layer could affect the drug sorption behavior. i.e., thickness, smoothness, net charge.

RE: Thank you for your remarks. It was added.

  1. Some minor points:

Line 363, it should be “0.5 – 5.0” not “0.5 ÷ 5.0”. Same for line 424;

RE: It was modified.

A high-resolution picture is suggested for Fig 3;

RE: It was modified.

In the Results and Discussion section, there are some descriptions regarding the techniques (i.e., the operation principle and scan parameter of SEM and FTIR.) that can be moved to the supplemental file or the previous Materials and Methods section.

RE: It was modified.

Thank you very much for your critical review. It was very useful in the correction of our manuscript. Identification of weak points throughout the text has helped us to increase the value of our paper. All comments and changes suggested by Reviewers have been incorporated into the manuscript. Once again, thank you very much for your help.

Round 2

Reviewer 2 Report

All the referres' comments have been satisfied and discussed.

Reviewer 3 Report

Authors have satisfied earlier issues. I recommend for accepting manuscript in the present form.